# Single-Dose Radiation Therapy for Localized Prostate Cancer: Where Does the Evidence Lead?

**DOI:** 10.3390/cancers17071176

**Published:** 2025-03-31

**Authors:** Salvatore Cozzi, Amina Lazrek, Giuseppe Rubini, Dino Rubini, Angela Sardaro, Sarah Houabes, Cecile Laude, Frederic Gassa, Lilia Bardoscia, Camille Roukoz

**Affiliations:** 1Radiation Oncology Department, Centre Leon Berard, 69373 Lyon, France; cecile.laude@lyon.unicancer.fr (C.L.); frederic.gassa@lyon.unicancer.fr (F.G.); 2Radiation Oncology Unit, International University Hospital Cheikh Zaid, Rabat 10000, Morocco; aminalazrek@yon.unicancer.fr; 3Section of Nuclear Medicine, Interdisciplinary Department of Medicine, University of Bari, 70124 Bari, Italy; giuseppe.rubini@uniba.it; 4Radiation Oncology Department, University of Bari “Aldo Moro”, 70124 Bari, Italy; dino.rubini@libero.it (D.R.); angela.sardaro@uniba.it (A.S.); 5Radiation Oncology Unit, Portes de Provence Hospital Groupe, 26200 Montélimar, France; sarah.houabes@ghpp.fr; 6Radiation Oncology Unit, S. Luca Hospital, Healthcare Company Tuscany Nord Ovest, 55100 Lucca, Italy; lilia.bardoscia@uslnordovest.toscana.it

**Keywords:** prostate cancer, radiotherapy, single fraction, hypofractionation, stereotactic body radiotherapy, brachytherapy

## Abstract

Radiotherapy treatment using modern techniques like stereotactic body RT is a safe and effective treatment for both localized and advanced prostate cancers. The progressive decrease in number of RT sessions from 40 to 20 and finally to five or four and perhaps one single fraction has been quite impressive. This manuscript reviews the evolution and current state of primary prostate single-dose radiotherapy, focusing on its benefits and limitations. Single-dose prostate RT approaches represent an exciting and potentially revolutionary opportunity in the field of RT treatment of localized prostate cancers. Despite some cases of disappointing long-term results from single-dose interventional brachytherapy, preliminary data from studies using stereotactic EBRT are extremely encouraging, especially in terms of toxicity.

## 1. Introduction

Prostate cancer (PCa) is the most prevalent cancer among men and the second leading cause of cancer-related mortality worldwide [1,2]. Most of the time, the diagnosis occurs when the cancer is still localized to the prostate gland, with a favorable prognosis, enabling various therapeutic options with curative intent, such as surgery, external beam radiotherapy (EBRT), brachytherapy (BT), and/or focal treatments. All these treatment options rely on advanced technologies and allow effective targeting of the tumor while minimizing the damage to the surrounding healthy tissues.

Recent evidence has confirmed that radiotherapy (RT) is a key treatment option for oligometastatic and oligorecurrent/oligoprogressive diseases, rare histological types, and in combination with new drugs for both hormone-sensitive and castration-resistant prostate cancer (PCa) [3,4,5]. Progress in RT planning and delivery methods has enhanced treatment precision and given rise to the adoption of hypofractionated radiation schedules even for the treatment of the prostate primary tumor [6]. International guidelines advocate the use of moderately hypofractionated RT schedules for the radical treatment of localized or locally advanced PCa as standard of care, since they have been fully confirmed through randomized trials as safe and effective compared to conventional fractionation [7,8,9,10,11,12,13]. In recent years, ultra-hypofractionated RT and proton beam therapy have emerged as promising RT techniques, with a number of ongoing studies assessing their efficacy and accessibility. In particular, stereotactic body radiation therapy (SBRT), delivering a high total radiation dose over a few RT sessions with high doses per session, has been associated with promising findings and an acceptable toxicity profile at the same time [5,14,15,16,17,18,19], given the relatively low alpha/beta value of prostate cancer cells increasing their sensitivity to high doses per fraction [20]. Nevertheless, cutting-edge technologies, i.e., MRI-guided RT, although still in the experimental stages, has been showing considerable potential for enhancing treatment outcomes and reducing side effects [21,22]. Low-dose rate brachytherapy (LDR-BT), that consists of a single session of total target high-dose intraprostatic seed implantation, is still proposed as an exclusive treatment option for early, localized, low- or intermediate-risk PCa, alternatively but equally effective for surgery or EBRT. High-dose rate brachytherapy (HDR-BT), usually in a single high-dose fraction, too, can be suggested for intermediate- or high-risk PCa, as a boost in association with EBRT to improve disease control and survival without worsening toxicity [6].

The most recent research strategy in the field of PCa definitive irradiation has been exploring prostate EBRT delivery in a single fraction. Such an approach could revolutionize the treatment of localized PCa, offering significant advantages from a biological, clinical, and cost-effectiveness point of view. The purpose of our article is to provide an overview of recent advances in the feasibility and delivery of single-dose RT (SDRT) for the treatment of localized PCa via EBRT or BT. By examining the progresses made thus far, we aim to highlight the potential benefits, challenges, and future perspectives of this treatment modality that moves among the consolidated data of interventional RT in the form of BT and the innovation of modern stereotactic radiation techniques.

## 2. Biological and Clinical–Dosimetric Rationale of Prostate Sdrt

Over time, advances in radiobiology have clarified how fractionation increases the therapeutic index of RT for many cancer types, preferentially causing malignant cell death over normal cells. DNA damage repair mechanisms influence the cytotoxic effects of radiation, and the differences in DNA repair capabilities between cancer cells and normal tissues enable ionizing radiation to target and eliminate tumor masses while minimizing harmful toxicity. As the dose per fraction decreases, a greater number of irradiated cells are able to repair sublethal DNA damage, resulting in an increased total dose needed to achieve a specific level of tumor cell death. Fractionation’s cell-sparing effect tends to be more significant for normal cells than for malignant cells, meaning that lower doses per fraction are more likely to protect normal tissues. This is the consolidated basis of the efficacy and safety of normofractionated RT regimens, for instance, in the treatment of head and neck cancers, needing a relatively long overall treatment time to achieve tumor-eradicating doses and an acceptable toxicity profile at the same time. The linear quadratic (LQ) model describes the dose–response curve of healthy and malignant tissues according to their alpha–beta ratio, with α/β characterizing the fractionation sensitivity of a particular cell type, depending on whether radiation-induced cell death is caused by lethal damage or the sum of sublethal damages. The α/β ratio is typically high (10 Gy) for tumor cells and relatively low (<5 Gy) for late-responding normal tissues. Tumor cells with a lower α/β than the surrounding organs at risk (OARs) theoretically are more sensitive to high doses per fractions, suggesting hypofractionation (that is, fewer but larger RT fractions) may be more effective in this case [23,24]. Given that the α/β ratio for PCa has been hypothesized to be nearly 1.5–3 Gy, the LQ model suggests that larger doses per fraction may result in an improved therapeutic ratio, together with selective OAR sparing as the bladder and rectum are known to have a higher α/β than PCa cells [25,26]. Building on such robust biological bases, several published trials regarding moderate hypofractionation for the treatment of PCa have shown similar curative rates than conventional fractionation (5 yr biochemical recurrence free survival (bRFS) 96.7% for low- and intermediate-risk disease and 92.1% for high-risk PCa) and similar late toxicity rates [7,8,9,10,22,27,28]. The encouraging data on moderate hypofractionation have given rise to the evaluation of the use of stereotactic treatments in the setting of prostate cancer RT, a sort of extreme form of hypofractionated RT usually delivered in four to seven fractions. The development and optimization of ultra-hypofractionation RT schedules over the last 20 years has resulted in the incorporation of SBRT into routine clinical practice, as a safe and cost-effective part of multidisciplinary, patient-tailored, therapeutic strategies. To date, ablative SBRT may be considered a standard of care for the treatment of various tumor types, including PCa, in the primary as well as in the recurrent and/or oligo-metastatic setting [29,30].

From a strict radiation oncology point of view, this was possible because SBRT has demonstrated similar dosimetric properties to high-dose-rate brachytherapy (HDR-BT) [31,32]. Likewise, BT dose distribution, which has been described as inversely proportional to the squared distance from the radiation source, thus owing to an optimal OARs sparing, the volumetric multiple-arcs intensity-modulated technique enables extremely inhomogeneous dose distributions, hence focusing very high RT doses within the tumor mass and simultaneously sparing the dose to the surrounding healthy tissues.

The beneficial effects of extreme hypofractionation may go beyond the delivered dose since prostate tumors have been observed to undergo genomic changes, including upregulation of genes responsible for immune regulation and apoptosis. Recent studies on the biology of SDRT have uncovered a distinct mechanism that differs fundamentally from the tumor cell-autonomous mechanism driving fractionated radiotherapy. At a threshold of 12 Gy, SDRT activates a dual-target model, combining acute tumor microvascular dysfunction with tumor cell DNA damage repair, resulting in synthetic tumor lethality [33]. In detail, SDRT causes acute vasoconstriction and hypoxic stress in tumor cells, prompting them to respond with increased mitochondrial production of reactive oxygen species. This leads to redox damage in the tumor cell nuclei, depleting chromatin-bound SUMO3, a crucial mediator for homology-directed repair of double-strand breaks (DSBs), such as BRCA1 and RAD51. While the extremely high doses used in clinical SDRT cause significant DSB damage, the loss of homology-directed repair leads to mis-repair or failure to repair DSBs, resulting in catastrophic tumor cell death and local tumor eradication. Clinical studies have confirmed that ablative SDRT induces acute microvascular perfusion and hypoxic defects, while hypofractionated SBRT does not cause microvascular dysfunction in tumor cells, even at ablative doses [33,34]. It is important to note that the LQ model may not accurately describe cell survival and isoeffects when the doses per fraction are ≥4 Gy. Indeed, the predictions of the LQ model for very high dose/fraction like SBRT and SDRT may either overestimate (less re-oxygenation, less sublethal damage repair, and thus an unavoidably lower “β” value) or underestimate (increased indirect cell death secondary to intravascular endothelial damage) the phenomenon [35,36].

## 3. Published Studies

### 3.1. The ONE-SHOT Study by Zilli et al. [37]

The ongoing ONE-SHOT study, a phase I/II trial, has been evaluating the safety and efficacy of prostate stereotactic SDRT at a dose of 19 Gy in patients with low- to intermediate-risk localized PCa.

The inclusion criteria were as follows: TNM stage cT1c-2c N0 M0; ISUP grade group 1–2; prostate-specific antigen (PSA) < 15 ng/mL; multiparametric magnetic resonance (mpMRI)-based prostate volume up to 70 cc; absence of significant tumors in the transitional zone and no extracapsular extension. Transurethral resection performed <12 weeks from RT was an exclusion criterion [37].

By using real-time verification and mitigation of prostate and OAR motion through electromagnetic transponders, the dose to the urethra has been reduced to 17 Gy. Preliminary results from the phase I segment of trial, involving six patients, have demonstrated no grade 2 gastrointestinal (GI) genitourinary (GU) adverse events in half of the patients over the first three months after the treatment. Zilli et al. also showed that the dosimetric impact of intrafraction prostate motion was minimal regarding target coverage using real-time electromagnetic tracking combined with beam gating. Recently, a trial update was published with the aim to evaluate the 3-year bRFS in 45 included patients. With a minimum follow-up of 18 months, no acute Grade 3 toxicity was recorded. The most severe genitourinary (GU) toxicity occurred on day 5 after SDRT, with 42.5% of patients experiencing grade 1 toxicity and 20% experiencing grade 2 toxicity, both returning to baseline by week 12. Late grade 2 toxicity was observed at month 18 in fewer than 3% of cases. Grade 3 proctitis with rectal bleeding occurred at month 12 in only one patient. The incidence of grade 2 or higher erectile dysfunction increased from 21.4% at baseline to 38.2% at month 18. Regarding quality of life (QoL), mean GU and gastrointestinal (GI) scores, as measured by the Expanded Prostate Cancer Index Composite (EPIC-26), showed a decline at day 5, complete recovery by month 6, and a new flare between months 12 and 18, with a consistent decline observed in the sexual domains. Overall, less than 20% of patients experienced a significant minimally clinically important change in EPIC GU and sexual scores. Mean prostate-specific antigen (PSA) values decreased from 6.8 ng/mL to 1.2 ng/mL at month 18 and further to 0.7 ng/mL at month 24 after SDRT. After a median follow-up of 26 months (range 24–36 months), three biochemical failures (7%) were recorded, all in patients with intermediate-risk prostate cancer (PCa) [37,38].

#### Treatment Planning and Radiation Delivery

All patients were treated with volumetric modulated arc therapy (VMAT) using 6–10 MV megavoltage beams with a flattening filter-free (FFF) modality. Additionally, all participating centers used Calypso^®^ (Varian Medical Systems, Palo Alto, CA, USA) electromagnetic transponders (beacons) implanted into the prostate at least one week prior to simulation and treatment planning for real-time image-guided radiation therapy (IGRT). Image verification was performed with cone beam computed tomography (CBCT) from the Linac system before, midway through, and at the end of each treatment.

Regarding the definition of volumes: the Clinical Tumor Volume (CTV) was delineated through co-registration with multiparametric MRI (mpMRI) as the prostate, with or without the proximal two-thirds of the seminal vesicles (SVs), depending on the risk of SV involvement determined by the Roach score, using a threshold cutoff of 15%. The Planning Target Volume (PTV) was defined as the CTV plus a 5 mm margin in all directions, except for a 3 mm margin posteriorly towards the rectal wall. The urethra Planning Risk Volume (PRV) was contoured on CT images by outlining a 12 French Foley catheter, with a 2 mm isotropic rim expansion. All organs at risk (OARs) were contoured according to RTOG guidelines, including the bladder, rectal walls (both defined with a 5 mm internal margin from the external surface), penile bulb, and proximal femurs [37,38].

### 3.2. The PROSINT Study by Greco et al. [39]

The phase II PROSINT study randomly assigned 30 intermediate-risk PCa patients to either 5 × 9 Gy prostate SBRT or 24 Gy SDRT to evaluate the toxicity profile, the PSA response, and patient-reported QoL measures. This study is the first and only randomized study comparing SDRT and SBRT. After a median follow-up of 48 months, SDRT exhibited a similar toxicity profile to fractionated SBRT. There was a peak of acute grade 1 GU symptoms one week after the treatment in both groups, including frequent urination and dysuria in 27% and 40% of SBRT and SDRT patients, respectively. Grade 1 side effects at three months were 7% for SBRT and 27% for SDRT, respectively. Cumulative actuarial late GU toxicities were comparable for grade 1 and grade ≥ 2. No grade ≥ 2 GI adverse events were observed, and no grade 4 treatment-related toxicity was reported. The median 3-year PSA level was 0.30 ng/mL for SBRT and 0.40 ng/mL in the SDRT arm, with both groups showing similar patient-reported outcomes in all the GI and GU domains.

PSA decline below 0.5 ng/mL occurred by 36 months in both arms. No biochemical relapses occurred in favorable intermediate-risk disease, while the unfavorable risk group showed an actuarial 4-year bRFS of 75.0% vs. 64.0% (HR, 0.76; 90% CI, 0.17–3.31) for SBRT vs. SDRT, respectively [39].

#### Treatment Planning and Radiation Delivery

Prior to the RT session, patients underwent a rectal enema and bladder voiding. An endorectal balloon was then inserted and inflated with 150 cm³ of air. A 12-French gauge Foley catheter, equipped with three embedded beacon transponders, was used for intrafractional tracking (Calypso, Varian Medical Systems). Fused computed tomography (CT) and T2-weighted three-dimensional MRI image sets were used to define the CTV, which included the prostate and the proximal two-thirds of the seminal vesicles, along with the OARs, including the rectal wall, bladder trigone, urogenital diaphragm, urethral wall, and neurovascular bundles. The PTV was defined as an isotropic 2 mm expansion of the CTV, with the margin reduced to 0 mm at the interface with the OARs. A 10-MV FFF beam energy and 4 VMAT arcs were used in all patients. Treatment delivery consisted in a linear accelerator with a 2.5 mm leaf width (EDGE, Varian Medical Systems). Daily CBCT matching ensured final target alignment, and online tracking-detected motion of ≥2 mm was realigned [40].

### 3.3. The ABRUPT Study by Arcangeli et al. [41]

In this single-arm prospective trial, 30 patients were enrolled to receive 24 Gy external beam SDRT to the whole prostate gland with urethra sparing and organ motion control.

The feasibility and safety of delivering single-dose 24 Gy in association with androgen deprivation therapy (ADT) in a cohort of patients with unfavorable intermediate and selected high-risk PCa patients were investigated. With a median follow-up of 18 months [6,7,8,9,10,11,12,13,14,15,16,17,18,19,20,21,22,23,24,25,26,27,28,29,30,31], no ≥G3 late adverse events were observed. The cumulative incidence of late GI and GU toxicity was 10.0% (3 patients) and 33.3% (10 patients), respectively. Late G2 bowel toxicity occurred in one (3.3%) patient, while two (6.7%) patients experienced G2 urinary toxicity. For late outcomes, Late GI toxicity of any grade was significantly associated with the maximum dose to the rectum (*p* = 0.021). The only statistically significant change among baseline, occurred 3 months after SDRT, and the last follow-up scores were observed in the urinary domain (*p* = 0.005). The median PSA level decreased from the baseline 8.05 ng/mL to 0.06 ng/mL after 3 months [41].

#### Treatment Planning and Delivery

All patients underwent a non-contrast-enhanced CT and mpMRI simulation. Anatomic reproducibility was ensured by administering a rectal microenema and bladder filling with 150 cc of saline solution using a 16-Foley catheter prior to both planning and treatment delivery. The CTV encompassed the entire prostate gland and seminal vesicles. The PTV consisted of the CTV plus an isotropic 2 mm margin. An isotropic 3 mm margin was also applied around the urethra, bladder, and rectum to determine the PRV. An initial CBCT soft tissue matching ensured accurate patient setup, and the real-time electromagnetic catheter-based prostate tracking system (Raypilot Micropos Medical AB) device was used to manage intrafraction organ motion [41].

### 3.4. The SiFEPI Study by Hannoun-Levi et al. [42]

The SiFEPI phase 2 clinical trial examined the efficacy and safety of HDR-BT with a single 20 Gy fraction in patients with low- and favorable intermediate-risk PCa. The study enrolled 35 patients between March 2014 and October 2017, with 33 patients completing the observation period. After a median follow-up of 72.8 months, the 6-year bRFS was 62%, the local relapse-free survival (lRFS) was 61%, and the metastasis-free survival (mRFS) 93%. The 6-year disease-free survival (DFS) rate was 54%, the cancer-specific survival (CSS) was 100%, and the overall survival (OS) was 89%. Late GU toxicity was observed in 33% of patients, GI in 12% of them, and sexual impairment in 21% of cases. Biochemical relapse was observed in 33% of patients, with a median time to relapse of 51 months, primarily due to local recurrences [42].

### 3.5. The Study by Salari et al. [43]

This is a prospective study on 26 men with low- or intermediate-risk PCa treated with a single 21 Gy fraction of prostate HDR-BT. With a median follow-up of 5.1 years, both grade ≥ 2 acute and late GU toxicity were observed in 38.5% of patients, with no grade ≥ 2 GI toxicity. The biochemical and local failure (LF) rates were 23.1% and 19.2%, respectively. No regional lymph node recurrences or distant metastases were reported. Five-year OS and cancer-specific survival (CSS) were 96.2% and 100%, respectively [43].

### 3.6. The Study by Hudson et al. [44]

A randomized phase II trial by Hudson et al. tested the efficacy of HDR-BT as monotherapy for localized PCa. Schedules consisting of 27 Gy in two fractions and a single fraction of 19 Gy were compared. The study included 170 patients with low- to intermediate-risk disease. After 8 years, the median PSA was 0.08 ng/mL in the two-fraction arm compared to 0.89 ng/mL in the single-fraction arm. The cumulative incidence of LF was significantly lower in the two-fraction arm (11.2%) versus the 19-Gy arm (35.9%) (*p* < 0.001). The rates of distant failure at 8 years were 3.8% in the two-fraction arm and 2.5% in the single-fraction arm, respectively (*p* = 0.6) [44].

### 3.7. The Study by Hoskin et al. [45]

This was a study comparing single-fraction HDR-BT (19 Gy or 20 Gy) to two-13-Gy-fraction and three-10.5-Gy-fraction interventional RT as exclusive treatment for intermediate- and high-risk localized PCa. After 10 years, the Kaplan–Meier estimates for biochemical relapse-free interval (bRFI) were 64% for the single-dose group, 72% for the two-fraction group, and 76% for the three-fraction group, respectively, with no statistically significant difference (*p* = 0.2). Multivariate analysis identified the risk class and ADT as significant predictors of relapse (*p* = 0.0003 and 0.03, respectively), while the delivered dose was not. Grade 3 GU events occurred in 8% of the single-dose group, 2% of the two-fraction group, and 13% of the three-fraction group (*p* = 0.01). An International Prostate Score Symptom (IPSS) ≥ 20 was found in 31% of the single-dose group, 20% of the two-fraction group, and 23% of the three-fraction group (*p* = 0.6). Grade 3 GI toxicity was 0% in the single-dose and two-fraction groups and 2% in the three-fraction group (*p* = 0.3). No grade 4 GU or GI events were recorded. Single-fraction HDR-BT also showed reduced, but not statistically significant differences in PSA control compared to multi-fraction regimens and OS differences [45].

### 3.8. The Study by Prada et al. [46]

Sixty consecutive patients with favorable clinically localized PCa were treated with HDR-BT as monotherapy, with one fraction of 19 Gy. Acute and late ≥grade 2 toxicity was not observed in any patients. The 6-year OS and failure in tumor-free survival (TFS) according to Kaplan–Meier estimates were 90% (±5%) and 88% (±5%), respectively. The 6-year actuarial biochemical control rate was 66% (±6%). The authors concluded that the protocol is feasible and very well tolerated, with low GU morbidity and no GI toxicity, but the obtained 6-year biochemical control was not as satisfactory as the same level of LDR-BT [46]. In view of such disappointing results in terms of local disease control, the Spanish group published the results of 60 patients treated with HDR-BT monotherapy using one higher fraction of 20.5 Gy. A six-year OS of 97% and failure in TFS of 98%, respectively, were reported in this case, with an actuarial biochemical control of 82%, and no ≥grade 2 acute and late toxicity were observed [47].

In their experience, a single dose of 20.5 Gy resulted in low GU morbidity and no GI toxicity, and it achieved better levels of biochemical disease control compared with one fraction of 19 Gy.

### 3.9. The Study by Siddiqui et al. [48]

This was a prospective trial of 19-Gy single-fraction HDR-BT for men with low- and intermediate-risk PCa. Sixty-eight patients were enrolled, with a median follow-up of 3.9 years. They reported a 14.7% incidence of late grade 2 GU toxicity, with no grade 3 urinary toxicity. One patient experienced grade 3 rectal toxicity (diarrhea), which was transient and resolved with medical management. The 5-year estimated DFS was 77.2%, with no significant difference based on the disease risk class. One patient developed distant metastases during the follow-up period. The biopsy-proven LF at 5 years was 18.8%, occurring at a median interval of 4.0 years post implant [48].

### 3.10. The Study by Tharmalingam et al. [49]

In 2020, the National UK HDR Prostate Brachytherapy Group published the results of a total of 441 patient series from seven UK hospitals treated with a single 19-Gy dose of HDR-BT. The median follow-up time was 26 months. The 3-year biochemical progression-free survival (bPFS) rates were 88% (overall), 100% (low-risk), 89% (favorable intermediate-risk), 81% (unfavorable intermediate-risk) and 75% (high-risk), respectively. Fifteen (3.4%) patients had a local, intraprostatic recurrence. Acute toxicity was generally mild, with no grade 4 events observed. The highest prevalence of acute grade 2 GU and GI toxicity occurred one month after the implant. Acute urinary retention that required catheterization was seen in 16 patients (5.4%). Late grade 3 GU toxicity developed in two patients (0.4%), both of whom required surgical treatment for urethral strictures. Additionally, two patients (0.4%) developed late grade 3 GI toxicity, both experiencing rectal fistulae that necessitated colostomy [49].

### 3.11. The Study by Morton et al. [50]

One hundred-eighty patients were accrued between June 2013 and April 2015 in this randomized, phase II clinical trial of prostate HDR-BT with one fraction of 19 Gy or two fractions of 13.5 Gy. Median follow-up was 60 months. The PSA decline was significantly faster in the two-fraction arm. The 5-year bPFS was 73.5% for the single-fraction arm and 95% for the two-fraction arm (*p* = 0.001). LF occurred in 23 (13%) patients in the single-fraction arm and 3 (2%) patients in the two-fraction arm, resulting in a 5-year cumulative LF incidence of 29.4% and 2.8%, respectively (*p* < 0.001). In the single-fraction arm, the stratified bPFS was 84%, 71%, and 68% for low, favorable intermediate, and unfavorable intermediate-risk groups, respectively, compared to 100%, 96%, and 93% in the two-fraction arm. In the single-fraction arm, local failure primarily occurred at the original site of disease within the prostate, whereas only one purely local recurrence was observed in the two-fraction arm. No significant differences in late toxicity were found between the two arms. Notably, there was very little GI toxicity, with only a 1% incidence of late grade 2 proctitis and no grade 3 GI toxicity. GU toxicity was more common, with 45% of patients experiencing grade 2 urinary retention. Only two (1%) patients had late grade 3 GU toxicity, both in the single-fraction arm. Grade 2 or higher erectile dysfunction (requiring medication) was observed in 45%, with no difference between the two arms [50].

Table 1 summarized the main point of aforementioned studies.

## 4. Discussion

The progressive decrease in the number of RT sessions from 40 to 20 and finally to 5 or 4 and perhaps 1 single fraction has been quite impressive.

Radiotherapy is essential in the treatment of PCa with curative intent. Recent advancements in radiation planning and delivery methods have enhanced treatment precision and led to the use of ultrahypofractionated radiation regimens, such as SBRT. The application of SBRT for localized PCa has demonstrated adequate tumor control, favorable patient-reported outcomes, and a low toxicity profile, making it a key component in the latest international guideline updates. Moreover, the COVID-19 pandemic has put great pressure on oncology services and admission, making ultrahypofractionation even more attractive. For instance, the shorter treatment duration facilitates a reduction in the risk of infection of patients and staff, with the net effect of an efficacy and tolerability profile comparable to longer, conventional treatment schedules. As a result, the international recommendations in response to COVID-19 for treating PCa already had indicated that five to seven stereotactic fractions should be used in centers with the appropriate technology [51,52]. This fascinating and surprising scenario fits into a complex, changing landscape in the field of prostate cancer diagnosis and treatment. The advent of next-generation imaging (mpMRI, prostate-specific membrane antigen–positron emission tomography (PSMA-PET/TC), genomic profiling, new drugs targeting the androgen receptor axis (Abiraterone, Enzalutamide, Apalutamide), but also molecular alterations in PCa cells (poly(ADP) ribose polymerase enzyme inhibitors) has been driving increasingly precise and patient-tailored treatments. Therefore, SDRT for the treatment of PCa is very attractive for every radiation oncologist, as it may simultaneously be cost-effective and patient-friendly. The single dose represents a kind of extreme form of ultrahypofractionation and might be useful to reduce the care burden without losing clinical efficacy, especially in view of the increase in PCa cases among elderly patients [53].

SDRT may offer non-negligible advantages from a biological and clinical but also cost-effectiveness point of view, including reduced hospital entrances that can be particularly beneficial for elderly patients or those living far from treatment centers. Moreover, it may provide increased convenience in theory, allowing patients to complete their treatment in a single session, a much more practical approach compared to standard regimens requiring not less than nineteen to twenty RT sessions. This reduces the overall treatment duration, improves patients’ QoL, and allows a relatively quick return to normal daily living activities. Fewer RT sessions may also enhance patient comfort by alleviating the emotional and physical burden associated with prolonged treatments.

### 4.1. Effective Dose for SDRT

Several reports have confirmed the feasibility and safety of single-dose RT using both BT and SBRT techniques, although the optimal target dose has not yet been established. The dose to be used to guarantee adequate local control of prostate cancer with a low toxicity profile represents the first key point to be clarified. Regarding brachytherapy, the study of Salari and colleagues suggested that exclusive, single-fraction 21-Gy HDR-BT is associated with a moderately high chronic urinary complication and significant local and biochemical failure rates, questioning its viability outside multimodal RT strategies in standard clinical practice [43]. Nevertheless, findings from the Hudson et al. randomized trial supported the use of HDR-BT delivered in two fractions of 13.5 Gy as a well-tolerated procedure that is able to provide sustained cancer control, while single-fraction monotherapy resulted in poor oncologic outcomes and should not recommended as exclusive treatment in the primary setting [44]. The data from the study by Prada et al. indicate that a dose escalation of more than 19 Gy is needed to improve biochemical control of disease in patients treated with a single fraction HDR-BT. Due to this evidence, the Spanish group then adopted a dose-escalation modality and provided better biochemical control without worsening toxicity in low and intermediate PCa risk by 20.5 Gy in one fraction of prostate HDR-BT. With a median follow-up of 51 months, morbidity was equal to that reported with the 19-Gy dose, while no ≥ grade 2 acute or late GU toxicity or any GI toxicity was recorded, and the 6-year actuarial biochemical control rate improved to 82% [47]. Hudson et al. demonstrated that a two-fraction HDR-BT regimen of 27 Gy resulted in significantly better cancer control compared to a single 19-Gy fraction, with lower LF rates and comparable OS, suggesting that it may be a more effective approach to deserve long-term disease control [44]. Such findings are in line with Tharmalingam et al. [49]. This multicenter study showed that a single dose of 19 Gy HDR-BT is safe and well tolerated for patients with low- and favorable intermediate-risk localized PCa, with good biochemical control rates over the first two years, while it looks to be insufficient for unfavorable intermediate- and high-risk patients. The high biochemical relapse rate and unexpected number of isolated intra-prostatic recurrences observed in the latter cohort supports the biological rationale for dose escalation in this setting by increasing the prescribed dose to the whole prostate gland or by using advanced imaging and planning techniques to deliver a focal boost to the dominant lesion [49]. Based on the LQ models and assuming an α/β ratio of 1.5 for the prostate tumor, BT schedules like 34.5 Gy in three fractions, 27 Gy in two fractions, or a single 19-Gy fraction would be expected to deliver biologically effective doses (BED) of 260–280 Gy (equivalent dose in 2 Gy per fraction (EQD2) 110–120 Gy). Sometimes, the high-risk PCa group may consist in rare histology variants, or aggressive, poorly differentiated tumor clones with higher α/β ratio than the acinar counterpart, or wider hypoxic areas; thus, they cannot respond to single large RT doses as well, likely due to the lack of re-oxygenation and temporal redistribution, compared to multi-fraction regimes. Morton et al., using MRI follow-up, demonstrated that recurrences were located almost exclusively at the site of the initial dominant disease [50]. All this considered, it appears unlikely that further single-fraction BT dose escalation would provide significant improvement in disease outcomes. Regarding EBRT, Italian and the Portuguese groups developed their own protocol using a single dose of 24 Gy on the entire prostate gland with a urethra-sparing, stereotactic, VMAT technique. The comparative analysis of ultra-high SDRT and extremely hypofractionated SBRT of the PROSINT study found similar PSA levels after 3 years, suggesting that 24 Gy SDRT could be a viable alternative to SBRT in four fractions, in line with the single-arm Italian ABRUPT Trial. However, the small sample size and relatively short follow-up duration of both the studies currently limit the generalization of these results [39,40,41]. To this day, there are no data concerning another local treatment in case of local recurrence. Would surgery or-re irradiation be an option without a high risk of complications?

### 4.2. Intrafraction Movement Control

Although the most mature data on prostate SDRT comes from BT experiences, such interventional RT modality presents a big question mark since it is an invasive technique, requiring general anesthesia, a high learning curve, and presents some unavoidable intrinsic perioperative risks. However, compared to EBRT, BT has the enormous advantage of the absence of intrafraction movement, which represents an important issue for external-beam SDRT. Both the Arcangeli et al. and Zilli et al. working groups have adopted advanced organ motion mitigation systems to ensure precise RT delivery [37,38,41].

The Calypso^®^ beacon (Varian Medical Systems, Palo Alto, CA, USA) system employs three electromagnetic transponders implanted in the prostate gland to assist treatment set-up and real-time target volume localization. A panel positioned above the patient contains a magnetic array that activates the beacons and captures the signals they emit. This enables real-time detection and continuous monitoring of the beacons’ positions relative to the treatment machine’s isocenter at a frequency of 10 Hz. The system uses the centroid of the three beacons to track the treatment target during dose delivery and identifies any changes in their positions during the patient positioning, to determine whether a beacon has shifted or significant deformation has occurred. While prostate rotation data are included in the treatment report, they are not utilized for target localization, because the treatment couch can only perform translational movements. Additionally, even slight beacon shifts or prostate deformation can create significant uncertainties in calculating the rotation angle, which the Calypso system cannot independently confirm [37]. Arcangeli and colleagues utilized a real-time electromagnetic (EM) tracking system incorporating a wired transmitter within a specialized lumen of the Raypilot HypoCath, a type of Foley catheter. The transmitter sends signals to the Raypilot Receiver, located beneath the patient on a carbon fiber treatment table. The transmitter comprises a choke coil (10 mm long and 3 mm in diameter) and a cable linked to the Raypilot Receiver before and during each treatment session to activate the device. The signals emitted by the transmitter are captured by an antenna array, allowing the system to calculate the transmitter’s position. Calibrated to the treatment room’s isocenter, the Raypilot system aids in both treatment localization and motion tracking. The software provides prostate position data along three-dimensional axes at a 30 Hz sampling frequency and also measures yaw (vertical axis rotations) and pitch (lateral axis rotations) [41].

### 4.3. Urethra Sparing

The urethra has been documented as an OAR potentially influencing the long-term toxicity of patients treated with definitive RT. A metanalysis of 23 SBRT prospective trials demonstrated a significant association between urethral doses and the onset of late GU toxicity, with each increase in 1 Gy in maximal urethral doses corresponding to a 0.8% and 1% increase in acute and late grade ≥ 2 GU toxicity, respectively. According to this model, a maximum urethral dose (Dmax) of 100 GyEQD2 would result in a 10% probability to experience late ≥grade 2 GU toxicity [54]. While strategies to reduce dose with strict urethral constraints may be effective when treating the prostate with a uniform dose and the dominant tumor not near the transition zone, urethra steering may be the preferred approach when dose escalation is required.

Several retrospective studies on HDR-BT have indicated that the bulbar–membranous urethra is the most radiosensitive section of the organ and is the most commonly affected by post-radiation stenosis. Mohammed et al. demonstrated a significant correlation between the risk of developing a urethral stricture and the Dmax delivered to the bulbo-membranous urethra [55,56,57].

Greco et al. found a trend towards increased patient-reported mild GU discomfort with SDRT and, similarly, a higher incidence of a transient late urinary flare syndrome following SDRT. They proposed that this temporary condition is likely caused by the dose delivered to the bladder trigone rather than the urethra, emphasizing the need for stricter dose/volume constraints in this area [39].

The urethra dose constraints and OARs constraints for each reported study are summarized in Table 2.

Accurate urethra delineation remains a challenge in clinical practice. While the use of a Foley catheter is substantially the standard method for the identification of the urethra, its invasive nature and the potential for treatment plan uncertainties due to urethral displacement limit its broader adoption. The adoption of mpMRI with dedicated imaging sequences and artificial intelligence (AI)-based automatic segmentation are promising tools to enhance urethral definition accuracy. Beyond the precise urethral definition, the advantages of this technology include adaptive treatment delivery with smaller PTV margins, compared to CT-guided SBRT techniques, and the ability to optimize for other healthy tissues associated with GU toxicity, such as the bladder trigone and neck, as well as the bulbar and membranous urethra.

## 5. Conclusions

Single-dose prostate RT approaches represent an exciting and potentially revolutionary opportunity in the field of RT treatment of localized PCa, offering reduced treatment duration and increased convenience with a very promising efficacy and tolerability profile. Despite some cases of disappointing long-term results from single-dose interventional BT, preliminary data from studies using stereotactic EBRT are extremely encouraging, especially in terms of toxicity. Appropriate high-precision targeting technology and advanced systems of target immobilization throughout treatment delivery are absolutely necessary to perform safe and cost-effective prostate SDRT. The added evidence of prostate volume shrinkage and cellular atrophy in post-treatment re-biopsies make SDRT comparable to a non-invasive, virtual prostatectomy. 

## Figures and Tables

**Table 1 cancers-17-01176-t001:** Summary table of the studies on single-dose treatment for PCa.

Trial Name	Phase	Patient Population	Treatment Regimen	Key Findings	Follow Up	Safety
ONE-SHOT [37,38]	Prospective Phase I/II	Low to intermediate-risk PCa	EBRT: 19 Gy SDRT (urethra dose reduced to 17 Gy)	Feasibility and tolerability demonstrated; 0% grade 2 GI toxicity 50% grade 2 urinary toxicity	18 months	Safe with manageable toxicity
PROSINT [39]	Prospective Phase II	Intermediate-risk PCa	EBRT: 5 × 9 Gy SBRT vs. 24 Gy SDRT	Similar toxicity profiles between SDRT and SBRT; 27% 1-week acute grade 1 GU symptoms (SBRT) vs. 40% (SDRT) 0% grade ≥ 2 GI toxicity	48 months	Comparable late GU toxicity; no grade 4 toxicity; PSA levels similar at 3 years
ABRUPT [41]	Single-arm prospective trial	Intermediate-risk PCa	EBRT: 24 Gy SDRT with urethra-sparing	Feasibility and safety demonstrated; 0% ≥ G3 late side effects	18 months.	Lower baseline QoL score, higher baseline IPSS score, acute GU toxicity, and acute urinary domain MID predicted GU toxicity of any grade
SiFEPI [42]	Prospective Phase II	Low-risk to favorable intermediate-risk PCa	HDR-BT: 1 fraction of 20 Gy	6y-bRFS 62% 6y-lRFS 61% 6y-mRFS 93% 6y-DFS 54% 6y-CSS 100% 6y-OS 89% 33% GU toxicity 12% GI toxicity 21% sexual impairment	72.8 months	Suboptimal biochemical control but positive late toxicity outcomes
Study by Salari et al. [43]	Prospective	Low to intermediate-risk PCa	HDR-BT: 1 fraction of 21 Gy	Moderate chronic urinary toxicity; significant local and biochemical failure rates	5.1 years	Single-fraction 21 Gy HDR associated with high chronic toxicity and failure rates, questioning its standard use
Study by Hudson et al. [44]	Prospective Phase II	Low to intermediate-risk PCa	HDR-BT: 27 Gy in 2 fractions vs. 19 Gy in 1 fraction	Two-fraction regimen shows better cancer control and lower failure rates; single-fraction has poorer outcomes	9 years	Two-fraction regimen provided sustained disease control Single-fraction less effective and not recommended
Study by Hoskin et al. [45]	Prospective Phase II	Intermediate- to high-risk PCa	HDR-BT: 19 Gy or 20 Gy single-dose vs. 13 Gy in 2 fractions vs. 10.5 Gy in 3 fractions	No significant difference in biochemical relapse-free interval; higher GU toxicity in single-dose group	10 years	Reduced PSA control with Single-dose HDR Multi-fraction regimens better disease control with acceptable safety
Study by Prada et al. [46]	Retrospective	Localized prostate cancer	HDR-BT: 1 fraction of 19 Gy	90% OS, 88% tumor-free survival at 6 years, 66% biochemical control at 6 years	72 months	No intraoperative or perioperative complications; no grade 2 or higher acute or late genitourinary toxicity; low GU morbidity; no GI toxicity
Study by Prada et al. [47]	Retrospective	low- and intermediate-risk prostate cancer	HDR-BT: 1 fraction of 20.5 Gy	Achieved good biochemical control with low GU morbidity and no GI toxicity	51 months	No intraoperative or perioperative complications; low GU morbidity; no GI toxicity; the protocol is feasible, well tolerated, and shows biochemical benefits at the dose of 20.5 Gy
Study by Siddiqui et al. [48]	Prospective trial Phase 2	low- and intermediate-risk prostate cancer	HDR-BT: 1 fraction of 19 Gy	5-year estimated DFS was 77.2%. Higher-than-expected rates of biochemical and LF. LF at 5 years was 18.8%.	3.9 years	Chronic grade 2 GU toxicity: 14.7%.; no grade 3 urinary toxicity; single case of grade 3 diarrhea (rectal toxicity) that resolved with medical management; single patient developed distant metastases; no deaths during follow-up
Study by Tharmalingam et al. [49]	retrospective	localized prostate cancer (all risk groups)	HDR-BT: 1 fraction of 19 Gy	3-year bRFS: 100% in low risk, 86% in intermediate risk, 75% in high risk; relapse predominantly occurred in the prostate; Gleason score was an independent predictor of bRFS	26 months	Acute toxicity was low with no grade 3 or 4 events. Two cases of late urinary stricture and two grade 3 late rectal events were reported. ADT was administered to 37.6% overall, with 90% of high-risk patients receiving it for a median of 6 months
Study by Morton et al. [50]	Prospective trial Phase 2	Low or intermediate risk prostate cancer, prostate	HDR-BT: 1 fraction of 19 Gy vs. 2 fractions of 13.5 Gy	5-year bRFS: 73.5% (single fraction) vs. 95% (two fractions); local failure: 29% (single fraction) vs. 3% (two fractions)	5 years	Urinary toxicity: Grade 2 (45%), Grade 3 (1%); late rectal toxicity: Grade 2 (1%); single fraction is inferior in cancer control; two fractions are well tolerated with higher efficacy

Abbreviations: pCa: prostate cancer; HDR-BT: high-dose rate brachytherapy; GU: genitourinary; GI: gastrointestinal; Gy: Gray; bRFS: biochemical recurrence free survival; LF: local failure; DFS: disease-free survival; OS: overall survival; SDRT: single-dose radiotherapy; EBRT: external beam radiotherapy.

**Table 2 cancers-17-01176-t002:** Dose constraints published in single-dose studies.

Study Name	Treatment Regimen	Dose Constraints
ONE-SHOT [37,38]	EBRT: 19 Gy SDRT	Urethra: Dmax < 17Gy
PROSINT [39]	EBRT: 5 × 9 Gy SBRT vs. 24 Gy SDRT	Urethral wall: D1 cm^3^ ≤ 19.2 Gy Bladder: D2% < 20.1 Gy Rectal wall: D5% ≤ 21.6 Gy; D50% ≤ 12.0 Gy; D1 cc ≤ 19.2 Gy Penile bulb: D2% < 19.2 Gy; D1 cm ≤ 12.0 Gy Femurs: D2% ≤ 12.0 Gy
ABRUPT [41]	EBRT: 24 Gy SDRT with urethra-sparing	Urethra PRV: D0.035 cc ≤ 22.8 Gy; D1 cc ≤ 19.2 Gy Bladder PRV: D0.035 cc ≤ 24.0 Gy; D1 cc ≤ 22.8 Gy; D50% ≤ 12.0 Gy Rectum PRV: D0.035 cc ≤ 22.8 Gy; D1 cc ≤ 19.2 Gy; D50% ≤ 12.0 Gy Neurovascular bundles: D0.035 cc ≤ 24.0 Gy; <50% of prescribed dose; ≤12.0 Femoral head: D0.035 cc ≤ 16.0 Gy Penile bulb: D0.035 cc ≤ 24.0 Gy
SiFEPI [42]	HDR-BT: 1 fraction of 20 Gy	Rectum: V85R ≤ 1%, Urethra: V110U < 1%
Study by Salari et al. [43]	HDR-BT: 1 fraction of 21 Gy	NA
Study by Hudson et al. [44]	HDR-BT: 27 Gy in 2 fractions vs. 19 Gy in 1 fraction	Urethra: Dmax < 120%; D10 < 115%; Rectal: Dmax < 90%; V80 < 0.2 cc
Study by Hoskin et al. [45]	HDR-BT: 19 Gy or 20 Gy single-dose vs. 13 Gy in 2 fractions vs. 10.5 Gy in 3 fractions	19 Gy in 1 fraction Rectum D2 cc < 15 Gy; V100 < 100% Urethra D10 < 22 Gy; D30 < 20.8 Gy; V150: 0 cc 20 Gy in 1 fraction Rectum D2 cc 15 Gy; V100 < 100% Urethra D10 < 22 Gy; D30 < 20.8 Gy; V150 < 0 cc
Study by Prada et al. [46,47]	HDR-BT: 1 fraction of 19 Gy	Rectum (Calculated at the anterior edge of the TRUS probe) Dmax ≤ 75% of the prescription dose Urethra: Dmax ≤ 110%
Study by Siddiqui et al. [48]	HDR-BT: 1 fraction of 19 Gy	Urethra: V110 < 10% Rectum: Dmax < 72.5%
Study by Tharmalingam et al. [49]	HDR-BT: 1 fraction of 19 Gy	Rectum: D2cc were <15 Gy with a maximum of <19 Gy Urethra: D10 < 22 Gy and D30 < 20.8 Gy with no area receiving >28.5 Gy
Study by Morton et al. [50]	HDR-BT: 1 fraction of 19 Gy vs. 2 fractions of 13.5 Gy	Urethra: Dmax < 120%, D10 < 115% Rectum: Dmax = 90%, V80 < 0.2 cc

Abbreviations: NA: not available; Dmax: maximum dose at one point; TRUS: transrectal ultrasound.

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
