# Peer review of "Single-Dose Radiation Therapy for Localized Prostate Cancer: Where Does the Evidence Lead?"

_cancers, 2025, doi:10.3390/cancers17071176_

Round 1
Reviewer 1 Report
Comments and Suggestions for Authors
Prostate cancer (PCa) is the most common cancer among men and the second leading cause of cancer-related deaths globally. This manuscript by Cozzi et al. reviews the advancements in single-dose radiotherapy (SDRT) for PCa, highlighting its benefits, such as reduced treatment time and convenience, alongside its challenges, including managing organ motion and minimizing toxicity. While the study is of interest to our readers, several issues warrant further attention.
1. It is not recommended to use section titles such as “3. Published Studies:” and “I. ONE-SHOT Study (Zilli et al.)” as they are neither suitable nor engaging. Additionally, please revise the section and subsection titles to follow a consistent format, such as using Arabic numbers: “3. xx,” “3.1. xx,” and so on.
2. A total of 11 studies were included; however, the summary of these studies should be presented in a more refined and elegant manner. Additionally, it would be beneficial to explore whether more relevant studies can be incorporated into this analysis.
3. To enhance readability, it is recommended to summarize the limitations of primary prostate SDRT in a table.
4. Please review and correct the inputs in Table 1, such as “PPhase/Phase,” “KEY FIN KEY hey/Key findings,” and “Efficacy Y And Efficacy,” among others.
5. On page 388, please revise “… and patient-tailored treatments ().”
Author Response
Dear reaviewer,
Thank you for reviewing our paper and thank you for your suggestions.
We hope that the modfications that we did met your expectations.
- It is not recommended to use section titles such as “3. Published Studies:” and “I. ONE-SHOT Study (Zilli et al.)” as they are neither suitable nor engaging. Additionally, please revise the section and subsection titles to follow a consistent format, such as using Arabic numbers: “3. xx,” “3.1. xx,” and so on. : We fixed the numbering of the sections and subsections. Concerning the first part of this comment, we are not sure to understand completely, we changed the first paragraph's name to : "The ONE-SHOT Study by Zilli et al." (we changed the others in the same way) .. Is this modification what you asked for?
- A total of 11 studies were included; however, the summary of these studies should be presented in a more refined and elegant manner. Additionally, it would be beneficial to explore whether more relevant studies can be incorporated into this analysis. : We decided to only write a narrative review about single dose prostate RT. Since there isnt a high number of published papers about this subject, we decided to talk about the most important ones.
- To enhance readability, it is recommended to summarize the limitations of primary prostate SDRT in a table.
We appreciate the reviewer’s suggestion to summarize the limitations of primary prostate SDRT in a table to enhance readability. While we understand the value of such an approach, we believe that the narrative discussion in the manuscript adequately addresses these limitations with the necessary detail and context. Summarizing this information in a table might oversimplify complex points, potentially reducing clarity and nuance.
We have carefully considered this suggestion and feel that retaining the detailed discussion in the text better serves the purpose of comprehensively addressing the limitations. Thank you for your understanding.
- Please review and correct the inputs in Table 1, such as “PPhase/Phase,” “KEY FIN KEY hey/Key findings,” and “Efficacy Y And Efficacy,” among others. : Thank you. That was fixed.
- On page 388, please revise “… and patient-tailored treatments ().” : Thank you. The "()" got deleted.
Reviewer 2 Report
Comments and Suggestions for Authors
Dear Editor,
Dear Author,
I read and reviewed the present paper submitted to cancers.
Single-Dose Radiation Therapy for Localized Prostate Cancer: Where does the evidence lead?
It is an interesting review article with a recapitulation on radiation therapy in prostate cancer. The authors provide deep and profound information. In my opinion the manuscript should be published. It contains important facts for clinicians.
However, I have some comments:
1. I am wondering if there are quality of life data related to the presented therapeutic modalities.
2. I would of benefit to discuss the results in context of the robotic surgery in prostate cancer.
3. I would speculate more about alternatives in patients in who the therapy fails?
4. The article is well written and it contains important information. Thanks for your efforts.
Author Response
Dear reviewer,
Thank you for reviewing our paper
- I am wondering if there are quality of life data related to the presented therapeutic modalities : Unfortunately, to this day, there is no data concerning the quality of life in patients treated with single dose RT.
- I would of benefit to discuss the results in context of the robotic surgery in prostate cancer.
We appreciate the reviewer’s suggestion to discuss the results in the context of robotic surgery in prostate cancer. While we acknowledge the growing role of robotic surgery in prostate cancer management, the primary focus of our study is on SDRT outcomes. Including a detailed discussion on robotic surgery may divert from the central objectives of the manuscript.
We have therefore decided to maintain our focus on SDRT to ensure clarity and coherence of the narrative. We hope the reviewer understands our decision to keep the scope of the discussion aligned with the study's aims.
- I would speculate more about alternatives in patients in who the therapy fails? Thank you, we added these sentences : "To this day, there is no data concerning another local treatment in case of local recurrence. Would surgery or re irradiation be an option without a high risk of complications?"
- The article is well written and it contains important information. Thanks for your efforts. : Thank you for your comments